# A global viral oceanography database (gVOD)

Le Xie[1], Wei Wei[1], Lanlan Cai[2], Xiaowei Chen[1], Yuhong Huang[1], Nianzhi Jiao[1], Rui Zhang[1, *], Ya-Wei Luo[1, *]

[1]State Key Laboratory of Marine Environmental Science, College of Ocean and Earth Sciences, Xiamen University, Xiamen, Fujian 361102, China

[2]Department of Ocean Science, The Hong Kong University of Science and Technology, Clear Water Bay, Hong Kong, China

*Correspondence to*: Rui Zhang (ruizhang@xmu.edu.cn) and Ya-Wei Luo (ywluo@xmu.edu.cn)

**Abstract.** Virioplankton are a key component of the marine biosphere in maintaining diversity of microorganisms and stabilizing ecosystems. They also contribute greatly to nutrient cycles/cycling by releasing organic matter after lysis of hosts. In this study, we constructed the first global viral oceanography database (gVOD) by collecting 10,931 viral abundance (VA) data and 727 viral production (VP) data, along with host and relevant oceanographic parameters when available. Most VA data were obtained in the North Atlantic (32%) and North Pacific Oceans (29%), while the Southeast Pacific and Indian Oceans were quite under sampled. The VA in the global ocean was $1.17(\pm3.31) \times 10^7$ particles ml$^{-1}$. The lytic and lysogenic VP in the global ocean was $9.87(\pm24.16) \times 10^5$ and $2.53(\pm8.64) \times 10^5$ particles ml$^{-1}$ h$^{-1}$, respectively. Average VA in coastal oceans was higher than that in surface open oceans [$3.61(\pm6.30) \times 10^7$ versus $0.73(\pm1.24) \times 10^7$ particles ml$^{-1}$], while average VP in coastal and surface open oceans was close. Vertically, VA, lytic and lysogenic VP deceased from surface to deep oceans by about one order of magnitude. The total number of viruses in the global ocean estimated by bin-averaging and the random forest method was $1.56 \times 10^{30}$ particles and $1.49 \times 10^{30}$ particles, leading to an estimate of global ocean viral biomass at 35.9 and 34.4 Tg C, respectively. We expect that the gVOD will be a fundamental and very useful database for laboratory, field and modelling studies in marine ecology and biogeochemistry. The full gVOD database (Xie et al., 2020) is stored in PANGAEA (https://doi.org/10.1594/PANGAEA.915758).

## 1 Introduction

Virioplankton are the most abundant biological entities and one of the largest genetic reservoirs in the ocean (Breitbart, 2012; Fuhrman, 1999). With an estimation of ~$10^{23}$ marine microbes being infected every second, viruses play important roles in affecting microbial mortality, regulating community composition and impacting biogeochemical cycles (Suttle, 2005; Zhang et al., 2007). Viruses were estimated to kill ~20–40% of marine bacterioplankton every day, a rate similar to that caused by zooplankton grazing (Fuhrman, 1999). In particular, virus-mediated cell lysis effectively 'shunts' approximately 25% of the photosynthetically fixed carbon, which otherwise would be transferred to higher trophic levels, to the dissolved organic matter (DOM) pool, partly forming the basis of the microbial loop and leading to the recycling of nutrients (Suttle, 2007; Wilhelm and Suttle, 1999). Furthermore, viral lysis can contribute to biological pump through the release of sticky lysates that accelerate the aggregation and sink of carbon into the deep sea (Suttle, 2005).

Compilation of the observations of viral abundance and activity in the global ocean is very necessary and urgent in understanding spatiotemporal distributions of viruses, exploring the controlling factors of viral processes, qualitatively and quantitatively assessing virus-host interactions and viral functioning in marine ecosystems, and even improving predictions of large-scale marine ecosystem and Earth system models. Previous two studies (Bar-On and Milo, 2019; Wigington et al., 2016) summarized viral abundance data in the ocean and estimated viral biomass as well as virus-to-prokaryote ratio. However, the lacking of host parameters such as bacterial production, and oceanographic parameters such as temperature, salinity, nutrient concentrations, limits the usage of these datasets in broader oceanographic contexts. More importantly, there is no public database of viral activity in the global ocean, which substantially hinders our understandings of the ecological and

biogeochemical roles of virioplankton on global scale. In addition, the ecological functions of the viruses are tightly linked to their life strategies, mainly including the lytic and the lysogenic infection (Wommack and Colwell, 2000). The significance of viruses in oceanic biogeochemistry is mainly reflected through the lytic infection, which results in cell lysis and the release of DOM. In contrast, other more temperate viruses choosing the lysogenic infection can influence microbial diversity and metabolism by transferring new genes to their hosts, altering the expression of host genes, and not killing hosts for many generations until an environmental or cellular trigger causes them to enter the lytic cycle. The lysogenic infection hence serves as a molecular "time bomb" (Paul, 2008). Therefore, it is necessary to include the quantity and quality data of viral life strategies in a viral oceanographic database.

In this study, we construct the first global viral oceanography database, namely gVOD, by collecting data of viral abundance (VA), lytic and lysogenic viral production (VP), as well as other related viral, host and oceanographic metadata when available. Based on the database, we estimate the total viral number and biomass in the global ocean. In addition, the data of VA and VP generated with different techniques were compared to provide references for evaluating possible technical biases.

## 2 Data and methods

### 2.1 Database summary

In the gVOD, direct measurements of three core parameters (VA, lytic and lysogenic VP), as well as accessary viral, prokaryotic and oceanographic parameters when available, were collected from published papers or acquired from lead authors or principal investigators (Table 1). Sampling information including date, latitude, longitude, depth and methods was included for each data record. We used ocean depth shallower or deeper than 200 m as a criterion to identify coastal or open ocean samples. The open ocean samples were further separated into surface and deep samples that collected in 0–200 m and >200 m, respectively.

The quality-controlled database consists of 10,931 VA data points (Appendix Table A1), 608 lytic VP data points and 119 lysogenic VP data points (Appendix Table A2). Most of VA (99.2%) and lytic VP (98.4%) and all lysogenic VP data have accompanying data of prokaryotic abundance (Table 1). For some samples, the abundances of flagellate, picoeukaryotes, *Synechococcus* and *Prochlorococcus* are also available. Prokaryotic productivity measurements cover 22.1% of VA, 57.7% of lytic VP and 76.5% of lysogenic VP data. The most available environmental parameters are salinity and temperature, providing oceanographic information for about half of VA, two-thirds of lytic VP and nearly all lysogenic VP data. Oxygen and chlorophyll *a* concentration data are also adequate particularly for VA. The concentrations of different types of nutrient, including nitrate, silicate and phosphate, are available for many samples. Other environmental parameters (pH, light intensity, dissolved organic carbon concentration) are relatively scarce. Moreover, given that the frequency of viral infected cells was calculated, independently or together with VP, usually to quantify the impact of viral infection within the microbial community (Chen et al., 2019; Payet and Suttle, 2013; Weinbauer et al., 2003), the reported frequencies of lytic infection (n = 438) and lysogenic infection (n = 266) in the literature were also collected into the database to facilitate the future exploration of marine

viral activities. Lastly, we collected 83 viral decay rate data, 206 viral burst size data and 111 virus-mediated mortality data, which can be useful for certain studies. The gVOD is a compilation of all the available data, to our best knowledge, by 2019.

We plan to update the database every 5 years.

## 2.2 Viral abundance

The viral abundance in this database was counted using one of the following three methods. In the first method, viruses were harvested by ultracentrifuging onto copper grids and stained with uranyl acetate, and then enumerated using a transmission election microscopy (TEM) (Akaike, 1974). In the second method, viruses were collected onto 0.02-μm filters and stained

with a nucleic acid-specific fluorescent dye (e.g., SYBR Green I), and then were counted under an epifluorescence microscope (EFM) (Noble and Fuhrman, 1998). The third method counted viruses by using flow cytometers (FCM), before which viruses were stained with fluorescent dye (e.g., SYBR Green I or SYBR Gold), and identified on the basis of the green fluorescence versus side scatter signal (Brussaard, 2004; Marie et al., 1999). The details of these three approaches have been described elsewhere (Weinbauer, 2004).

## 2.3 Lytic viral activity

Lytic VP is paramount and widely employed to assess the activity of lytic viruses in community-level and the roles of viruses in marine ecosystems. In this database, the lytic VP was estimated by one of the following five methods. The first method estimated VP by calculating expected viral release rates by multiplying fraction of viral infected cells (mainly prokaryotes), prokaryotic productivity (assuming equal prokaryotic mortality rate) and burst size obtained from TEM studies (Noble and

Steward, 2001) or virus-dilution approach (Weinbauer et al., 2002). For notational simplicity, in this paper we label this method as FPB to represent the three variables (Fraction of viral infected cells, Prokaryotic productivity, and Burst size) listed above and used in the estimation. In the second method, called radioactive incorporation approach (RIA), lytic VP was estimated by determining viral DNA synthesis rates using a labelled radiotracer (e.g., $^3$H-, $^{32}$P-, or $^{14}$C-labeled thymidine or leucine) and a conversion factor to quantify the incorporated radiotracer into viral particles (Noble and Steward, 2001; Zimina et al., 1973).

The third method estimated the lytic VP from the viral decay rates (VDR), assuming that the abundance of virus particles is in steady state and then the loss rate of virus particles should be balanced by the production rate (Heldal and Bratbak, 1991). The fourth method used fluorescently labelled viral tracers (FLVT) to measure the dilution rates from the decay of labelled viruses and net changes of the non-labelled viruses in natural viral community (Noble and Fuhrman, 2000). The fifth method quantified the increase of viral abundance during time course incubation using a virus dilution or virus reduction approach (VRA)

(Weinbauer et al., 2010; Winget et al., 2005), which effectively avoided new viral infection by reducing viral abundances using pore-size filters or tangential flow filtration systems.

## 2.4 Lysogenic viral activity

Lysogenic VP is generally measured by detecting the proviruses (temperate viruses) that choose lysogenic infection in the environment. Lysogenic VP in this database was estimated using VRA described above after the provirus induction by Mitomycin C (Weinbauer et al., 2002). Hence, the lysogenic VP was estimated as the difference in viral abundance per unit time between the Mitomycin C treated and the control samples.

## 2.5 Quality control

We conducted quality control for the VA, lytic and lysogenic VP data of the database. A negative lytic VP (Wells and Deming, 2006) was removed. All zero-value (below detection limit) data were kept in the database but were not included in the following analyses. For those positive-value data, we applied the Chauvenet's criterion to their log-transformed values to identify outliers (Glover et al., 2011): A datum was treated as an outlier when its probability of deviating from the observed mean was lower than $1/(2n)$, where n was the number of data samples. Outliers were marked in the database and not included in the following analyses.

## 2.6 Total number and biomass of viruses in global ocean

Based on the VA data of our database, we estimated the total number of viruses in the global ocean using two methods. In the first method, we separately estimated the total number of viruses in different ocean basins, including the North and South Atlantic Ocean, the North and South Pacific Ocean, the Indian Ocean, the Arctic Ocean, and the Mediterranean and Baltic Seas. For each basin, viral numbers were calculated for coastal and open oceans separately. The mean VA in each depth bin (coastal oceans: bins separated at 5, 10, 25, 50 and 100 m; open oceans: bins separated at 10, 20, 30, 40, 50, 60, 70, 80, 90, 100, 200, 500, 1000 and 2000 m) were multiplied by its water volume to calculate the number of viruses in that bin. Please note that due to the insufficient data, mean VA in deep waters of the Arctic Ocean, Mediterranean Sea and Baltic Sea were substituted by the value of the North Atlantic Ocean. The total number of viruses in global ocean was then calculated by summing up the estimates of the coastal and open ocean regions of all the counted ocean basins.

The second method used the random forest (MATLAB machine learning toolbox) (Breiman, 2001) to construct a model of VA based on sampling latitude, longitude, months and depths. VA data were binned to $1° \times 1°$ with 44 vertical layers and the mean VA of each bin, if data available, was fed into the random forest. When implementing the random forest, 75% samples were randomly selected for training the model while the rest data were used for model validation. The trained model was then used to predict VA for each bin and then to estimate the total viral number in the global ocean.

The viral biomass of the global ocean was calculated from the virus numbers using a conversion factor of 0.023 fg C per viral particle, which was based on an empirical relationship between carbon contents in heads of marine viruses ($C_{head}$) and their sizes (Jover et al., 2014):

$$C_{head} = 41(r - 2.5)^3 + 130(7.5r^2 - 18.75r + 15.63), \tag{1}$$

where $r$ was the radium of viral head for which an average of 26.3 nm from the Tara Ocean expedition data was used (Brum et al., 2015).

In this paper, all the uncertainties reported in parentheses after the means are standard deviations, except that the standard errors of the mean are reported for the estimates of total viral number and biomass of the global ocean, because the mean values are used in the estimates and therefore the uncertainties of the means are the most interested.

## 3 Results and discussion

### 3.1 Data distribution

Most VA data were collected in the north hemisphere (particularly in tropical and subtropical regions), while fewer data in the southern hemisphere (Figs. 1a−1c). In total, nearly two-thirds of the VA data were sampled in the North Atlantic Ocean (32%) and North Pacific Ocean (29%) (Fig. 2a). In addition, 6 long-term time-series of VA were included in the compilation (Figs. 1a & 1b): the Bermuda Atlantic Time-series Study (BATS) in 2000–2009, the San Pedro Ocean Time Series (SPOTS) Microbial Observatory in 2000–2011, the Bedford Basin Monitor (BBM) in 1996–2000, the Rivers Inlet (RI) in 2008–2010,

the Saanich Inlet (SI) in 2010–2012 and 2014–2015, and the Guanabara Bay (GB) in 2011–2014. Weekly VA were measured at BBM, approximately monthly samples were collected at BATS, SPOTS, SI, and GB year-round, and monthly samples were collected at RI only in spring and summer. Vertically, most VA data were sampled in the surface ocean ($\leq 200$ m, 71%) while fewer data in the deep ocean ($> 200$ m, 29%) particularly below 1,000 m (Fig. 3a). Summer VA samples were most abundant while the fewest data in winter (Fig. 4a).

Lytic VP data in the north hemisphere are much more than those in the south hemisphere (Figs. 1d−1f), with almost half of lytic VP data were sampled in the North Pacific Ocean (31%) and North Atlantic Ocean (18%) (Fig. 2b). A majority of lytic VP data (86%) was collected in the surface ocean (Fig. 3b), while the deep samples were mostly from the North Atlantic and the western and northeastern Pacific Oceans (Fig. 1e). There were seasonal biases in lytic VP data, most of which were sampled in summer while rarely sampled in autumn (Fig. 4b). More lytic VP data were sampled in open oceans (63%) than in coastal

waters (37%) (Fig. 5). Almost every lytic and lysogenic VP data accompanied with VA measurements.

There were very limited number of lysogenic VP data in both surface and deep oceans (Figs. 1g & 1h), with those deep samples being even much fewer than the already scarce surface ones (Fig. 3c). The northern hemisphere had slightly more lysogenic VP data than the southern hemisphere (Fig. 1i), with most lysogenic VP data sampled in the North Pacific (29%), the North Atlantic (24%) and the South Atlantic Oceans (23%) (Fig. 2c). Similar to lytic VP data, lysogenic VP data were

tended to be collected in spring and summer than in other seasons particularly winter (Fig. 4c). Lysogenic VP data in the open ocean (77%) were also much more than those in coastal waters (23%) (Fig. 5).

In summary, most viral data were sampled in North Atlantic and Northeast Pacific Oceans (Figs. 1 & 2), and more data in the surface than in the deep oceans (Fig. 3). Viral data also tended to be sampled in summer (Fig. 4). Although the total viral

data in the coastal samples were fewer than the open ocean samples (Fig. 5), they were more concentrated in the coastal zones
considering their relatively small area in the global ocean.

## 3.2 Viral abundance in the global ocean

In the surface oceans, VA (n=7,768) mostly varied in the order of $10^6$ to $10^8$ particles $ml^{-1}$, with mean VA in coastal waters [3.61($\pm$6.3) $\times 10^7$ particles $ml^{-1}$] about 5 times higher than that in the open oceans [7.3($\pm$12.4) $\times 10^6$ particles $ml^{-1}$] (Figs. 6a & 6b). VA in coastal South Atlantic Ocean and Mediterranean and Baltic Seas was higher than that in other coastal oceans (Fig.
6a). Although the VA across different surface open oceans distributed in similar ranges, the average VA in Pacific (particularly in its southern portion) was higher than those in other basins (Fig. 6b), a pattern previously found in another study (Lara et al., 2017) using fewer data than this study. VA decreased with depth, with those in the global deep ocean [1.26($\pm$2.44) $\times 10^6$ particles $ml^{-1}$, n=3,164] about one order of magnitude lower than those in the surface (Figs. 7a & 7b). The vertical profiles in different open ocean basins more clearly showed that the VA in the Pacific was higher than that in the Atlantic in surface 1,000
m, while the difference did not exist in deeper oceans (Fig. 7b).

In our database, most VA samples were measured using FCM (7,353, 67.26%) and EFM (3,465, 31.71%), while only 112 (1.03%) VA samples were counted using TEM (Appendix Table A1). Previous studies have showed that the VA counted using FCM, which became more popular in studies after 2014 (Appendix Table A1), had a strong correlation with those using EFM (Brussaard et al., 2010; Marie et al., 1999; Payet and Suttle, 2008). Our data demonstrated that the VA obtained by FCM and
EFM methods has consistent results in similar environments. For deep open ocean samples, VA using TEM are substantially lower than those using the other two methods (Fig. 8). But considering much fewer VA data points using TEM than others (Fig. 8 & Appendix Table A1), we cannot conclude TEM substantially underestimated VA in the deep water samples. Nevertheless, our database provides references for methodological comparison in the future.

The total number of global ocean viruses estimated by binning the VA data (Figs. 7a & 7b) is 1.56($\pm$0.2) $\times 10^{30}$ particles
(mean$\pm$s.e.), which is very close to the estimate of 1.49($\pm$0.14) $\times 10^{30}$ particles (mean$\pm$s.e.) using the random forest method (Appendix Figure A1). Both values are consistent to the previous estimates of $10^{30}$ (Suttle, 2007), 1.29 $\times 10^{30}$ (Cobian Guemes et al., 2016) and 1.5 $\times 10^{30}$ (Bar-On and Milo, 2019) viral particles for the global ocean. Using a conversion factor of 0.023 fg C per viral particle (see Methods), our two values of total viral number give the estimates of total viral biomass in the global ocean at 35.9 $\pm$0.46 and 34.4 $\pm$0.32 Tg C, respectively, confirming a recent estimate of 30 Tg C (Bar-On and Milo, 2019).

## 3.3 Viral production

In surface ocean, lytic VP (n=523) varied greatly from $10^3$ to $10^7$ particles $ml^{-1} h^{-1}$ in different ocean basins (Figs. 6c & 6d). The overall mean and standard deviation of lytic VP in the global ocean were 9.87($\pm$24.16) $\times 10^5$ (ranging in 0.00746 $\times 10^5$ – 350 $\times 10^5$) particles $ml^{-1} h^{-1}$. Lytic VP values in surface open Pacific Ocean were about one order of magnitude higher than those in surface open Atlantic Ocean (Fig. 6d), a pattern consistent to VA (Fig. 6d). Lytic VP in the surface Arctic Ocean was
much lower than that in other basins, which was expected considering its much lower biological productivity (Figs. 6c & 6d).

Although insufficient lytic VP data (n=82) were available for meaningful statistical analyses in the deep waters (Fig. 9), the existing data showed a general trend that VP decreased by one order of magnitude from the surface to the deep open oceans (Fig. 7c). Unlike VA, average lytic VP in coastal waters was close to that in the surface open ocean (Fig. 6c).

Most of the lytic VP (84.4%) in this database was estimated by VRA, suggesting that VRA was widely utilized in literature and became a standard method to estimate VP across different marine environments. Several studies have tried to compare different approaches estimating the lytic VP, revealing that the VRA method was more reliable and less laborious, compared to the probable overestimation by FLVT approach and the potential underestimation by RIA method, though such comparisons were mainly constrained to the coastal ocean (Helton et al., 2005; Karuza et al., 2010; Rastelli et al., 2016; Winget et al., 2005). Additionally, although a meaningful comparison of reported lytic VP values between disparate marine ecosystems is complicated by the inherent variability among approaches, the lytic VP rates in this database might provide a tentative global-scale insight into methodological comparison. Our statistics showed that, in similar environments, the lytic VP rates determined by FLVT and VRA were higher than those measured by RIA. For coastal samples, such difference among methods was not obvious (Fig. 9). However, due to the limited number of samples using the methods other than VRA (Fig. 9 and Appendix Table A2), we did not have adequate data to tell if the difference in VP was caused by the measurement methods, or the randomness of the samples. Hence, more measurements of lytic VP using multiple approaches simultaneously will be certainly needed to better evaluate the differences among them.

The lysogenic VP data were too few (surface n=85, deep ocean n= 34) for meaningful comparisons across different ocean basins or between the surface and deep waters, although the results were plotted for readers' reference (Figs. 6e & 6f). The overall lysogenic VP in the global ocean was estimated at $2.53(\pm 8.64) \times 10^5$ (ranging in $0.00132 \times 10^5 - 68.8 \times 10^5$) particles ml$^{-1}$ h$^{-1}$, which was about one third of the level of lytic VP, although more data will be needed to better compare the two types of VP.

## 4 Data availability

The gVOD database (Xie et al., 2020) can be downloaded from PANGAEA at https://doi.org/10.1594/PANGAEA.915758.

## 5 Code availability

The MATLAB codes for calculating the total number of viruses can be found in the supplementary materials or be obtained by requesting the corresponding authors.

## 6 Conclusion

We constructed a global ocean viral database (gVOD) by compiling 10,931 VA data, 608 lytic VP data and 119 lysogenic VP data. This database may be useful for global-scale studies of viral processes and their roles in marine ecosystems and biogeochemical cycles. The VA, lytic and lysogenic VP data were greatly variable. Most VA were counted using flow cytometers and epifluorescence microscopes, while the virus reduction approach was the most popular method in estimating VP. The lytic VP is about 3 times higher than the lysogenic VP. The calculation using the database also confirms the previous estimates of viral numbers and biomass in the global ocean.

Our database shows that the current investigations have the limitation in spatiotemporal coverage. The VA dataset has a poor coverage in South Pacific and Indian Ocean. The lytic VP dataset does not have a good coverage in South Pacific, Northwest Pacific, Indian and South Atlantic Oceans. The lysogenic VP data are very few in the global ocean. Vertically, all viral data were sampled much less in mesopelagic and deep oceans than in the surface oceans. Thus, the measurements of viral parameters in these regions and depths should be given high priority. In addition, more viral data should be sampled in winter to avoid seasonal biases.

The database is stored in a public data repository (PANGAEA), and will be updated regularly when new data become available. We hope that the database will be valuable for field and modelling studies in marine ecology, biogeochemistry and other areas of oceanography.

**Appendix Tables and Figures**

**Appendix Table A1. Sources and methods of viral abundance data. EFM: counted by epifluorescence microscopes. FCM: counted by flow cytometer. TEM: counted by transmission election microscopy. Data marked by * are those collected in a previous dataset (Wigington et al., 2016).**

| Region | Number of data | Method | References |
|---|---|---|---|
| Canadian Arctic Shelf * | 259 | EFM | (Payet and Suttle, 2013) |
| Gulf of Alaska, Arctic * | 292 | EFM | (Balsom, 2003) |
| Franklin Bay (Arctic Ocean) | 4 | EFM | (Wells and Deming, 2006) |
| Greenland Sea, Arctic | 79 | FCM | (Boras et al., 2010a) |
| Greenland Sea * | 124 | EFM | (Wigington et al., 2016) |
| Arctic Ocean | 56 | FCM | (Finke et al., 2017) |
| Kora Sea (Arctic Ocean) | 18 | EFM | (Kopylov et al., 2019) |

| | | | |
|---|---|---|---|
| North Sea | 16 | EFM | (Weinbauer et al., 2002) |
| North Sea | 9 | EFM | (Winter et al., 2005) |
| North Sea | 39 | FCM | (Parada et al., 2008) |
| North Sea * | 191 | FCM | (Wigington et al., 2016) |
| North Sea * | 95 | FCM | (Wigington et al., 2016) |
| North Sea | 23 | FCM | (Winter et al., 2004) |
| North Atlantic | 5 | TEM | (Proctor and Fuhrman, 1990) |
| North Atlantic * | 188 | FCM | (Li and Dickie, 2001) |
| North Atlantic | 11 | EFM | (Auguet et al., 2005) |
| North Atlantic | 6 | FCM | (Parada et al., 2007) |
| North Atlantic | 20 | TEM | (Bettarel et al., 2008) |
| North Atlantic | 31 | EFM | (Rowe et al., 2008) |
| North Atlantic * | 772 | FCM | (De Corte et al., 2012) |
| North Atlantic | 9 | FCM | (Muck et al., 2014) |
| North Atlantic * | 206 | FCM | (Mojica et al., 2015) |
| North Atlantic | 41 | EFM | (Parsons et al., 2015) |
| North Atlantic | 10 | FCM | (Winter et al., 2018) |
| North Atlantic | 39 | FCM | (Finke et al., 2017) |
| Gulf of mexico | 14 | TEM | (Jiang and Paul, 1996) |
| Gulf of Mexico | 9 | EFM | (Weinbauer and Suttle, 1996) |
| Gulf of Mexico | 12 | TEM | (Cochran and Paul, 1998) |
| Gulf of Mexico | 7 | EFM | (Weinbauer and Suttle, 1999) |
| Gulf of Mexico | 25 | EFM | (Williamson et al., 2002) |
| Gulf of Mexico | 28 | EFM | (Long et al., 2008) |
| Chesapeake Bay | 7 | EFM | (Winget and Wommack, 2009) |
| Chesapeake Bay * | 84 | EFM | (Wang et al., 2011) |
| Sargasso Sea * | 1382 | EFM | (Parsons et al., 2012) |
| Tropical Atlantic | 10 | EFM | (Winter et al., 2008) |
| Tropical Atlantic | 154 | FCM | (De Corte et al., 2010) |
| Atlantic Ocean | 426 | FCM | (Lara et al., 2017) |

| | | | |
|---|---|---|---|
| South Atlantic | 3 | EFM | (Bettarel et al., 2011b) |
| South Atlantic | 172 | FCM | (Liang et al., 2014) |
| South Atlantic | 3 | FCM | (Gregoracci et al., 2015) |
| South Atlantic * | 430 | FCM | (De Corte et al., 2016) |
| Guanabana Bay (South Atlantic) | 246 | FCM | (Cabral et al., 2017) |
| Adriatic (Mediterranean) | 35 | TEM | (Weinbauer et al., 1993) |
| Mediterranean | 25 | EFM | (Bettarel et al., 2002) |
| Mediterranean and Baltic Sea | 30 | EFM | (Weinbauer et al., 2003) |
| Adriatic (Mediterranean) | 4 | EFM | (Bongiorni et al., 2005) |
| Mediterranean | 48 | EFM | (Magagnini et al., 2007) |
| Mediterranean | 24 | FCM | (Boras et al., 2009) |
| Mediterranean | 3 | EFM | (Motegi et al., 2009) |
| Mediterranean | 43 | FCM | (Winter et al., 2009) |
| Mediterranean | 9 | EFM | (Fonda Umani et al., 2010) |
| Mediterranean | 6 | FCM | (Bouvier and Maurice, 2011) |
| Mediterranean | 10 | EFM | (Maurice et al., 2011) |
| Mediterranean | 338 | FCM | (Magiopoulos and Pitta, 2012) |
| Mediterranean | 45 | EFM | (Maurice et al., 2013) |
| Mediterranean | 1 | FCM | (Motegi et al., 2014) |
| Mediterranean | 2 | FCM | (Thompson et al., 2014) |
| Mediterranean | 21 | EFM | (Ordulj et al., 2017) |
| Baltic Sea | 6 | FCM | (Holmfeldt et al., 2010) |
| Baltic sea | 9 | FCM | (Kostner et al., 2017) |
| Baltic sea | 4 | EFM | (Šulčius et al., 2018) |
| Indian Ocean * | 52 | EFM | (Wigington et al., 2016) |
| Indian Ocean | 93 | FCM | (Liang et al., 2014) |

| | | | |
|---|---|---|---|
| Cochin Estuary (Indian) | 35 | EFM | (Parvathi et al., 2013) |
| Cochin Estuary (Indian) | 20 | EFM | (Jasna et al., 2017) |
| Cochin Estuary (Indian) | 39 | EFM | (Jasna et al., 2018) |
| Indian Ocean | 271 | FCM | (Lara et al., 2017) |
| Indian Ocean | 33 | EFM | (Parvathi et al., 2018) |
| Red Sea | 51 | FCM | (Sabbagh et al., 2020) |
| Bering Sea | 12 | TEM | (Steward et al., 1996) |
| Bering Sea | 15 | FCM | (Finke et al., 2017) |
| Santa Monica Bay | 7 | EFM | (Noble and Fuhrman, 2000) |
| Japan sea | 12 | EFM | (Hwang and Cho, 2002) |
| Masan Bay | 24 | EFM | (Choi et al., 2003) |
| North Pacific | 36 | EFM | (Taylor et al., 2003) |
| North Pacific (HOT) | 8 | EFM | (Brum, 2005) |
| San Pedro Channel (North Pacific) | 386 | EFM | (Fuhrman et al., 2006) |
| North Pacific | 11 | EFM | (Hewson and Fuhrman, 2007) |
| Strait of Georgia (North Pacific) * | 67 | EFM | (Clasen et al., 2008) |
| Bach Dang Estuary (North Pacific) | 15 | TEM | (Bettarel et al., 2011a) |
| North Pacific * | 355 | FCM | (Yang et al., 2014) |
| Northwestern Pacific | 39 | FCM | (Li et al., 2014) |
| Pearl River Estuary (North Pacific) | 19 | FCM | (Ni et al., 2015) |
| North Pacific | 4 | EFM | (Pasulka et al., 2015) |
| Bohai sea | 8 | FCM | (Ma et al., 2016) |
| North Pacific * | 9 | EFM | (Wigington et al., 2016) |
| North Pacific | 399 | FCM | (Finke et al., 2017) |

| | | | |
|---|---|---|---|
| North Pacific | 37 | EFM | (Gainer et al., 2017) |
| Western Pacific | 244 | FCM | (Liang et al., 2017) |
| Western Pacific | 222 | FCM | (Zhao et al., 2020) |
| Jiulong Estuary (North Pacific) | 27 | FCM | (Chen et al., 2019) |
| South China Sea | 751 | FCM | unpublished |
| South China Sea | 6 | FCM | (Zhang et al., 2007) |
| South China Sea | 13 | FCM | (Chen et al., 2011) |
| South China Sea | 5 | EFM | (Nguyen-Kim et al., 2015) |
| Mariana Trench | 6 | FCM | (Li et al., 2018) |
| Pacific Ocean | 171 | FCM | (Liang et al., 2014) |
| Pacific Ocean | 331 | FCM | (Lara et al., 2017) |
| South Pacific * | 12 | EFM | (Wilhelm et al., 2003) |
| South Pacific | 31 | EFM | (Strzepek et al., 2005) |
| South Pacific | 2 | EFM | (Motegi C and Nagata T, 2007) |
| South Pacific * | 24 | EFM | (Rowe et al., 2012) |
| South Pacific | 10 | EFM | (Bouvy et al., 2012) |
| South Pacific * | 16 | EFM | (Matteson et al., 2012) |
| South Pacific * | 542 | FCM | (Yang et al., 2014) |
| Southern Ocean (Indian sector) | 14 | FCM | (Evans et al., 2009) |
| Southern Ocean (Indian sector) | 10 | FCM | (Malits et al., 2014) |
| Southern Ocean (Atlantic sector) * | 33 | FCM | (Evans and Brussaard, 2012) |
| Southern Ocean (Atlantic sector) | 25 | EFM | (Brum et al., 2016) |
| Southern Ocean (Pacific sector) | 161 | FCM | (Vaque et al., 2017) |

**Appendix Table A2. Sources and methods of VP data. FPB: Calculated by multiplying fraction of viral infected cells, prokaryotic production and burst size; RIA: radioactive incorporation approach; FLVT: fluorescently labeled viral tracers method; VRA: virus reduction approach (see main text section 2.3 for details). MC: Virus reduction approach with addition of inducing agent such as Mitomycin C (see main text section 2.4 for details).**

| Region | #lytic VP | Method of lytic VP | # lysogenic VP | Method of lysogenic VP | References |
|---|---|---|---|---|---|
| Canadian Arctic Shelf | 14 | VRA | - | - | (Payet and Suttle, 2013) |
| Franklin Bay (Arctic Ocean) | 3 | VRA | - | - | (Wells and Deming, 2006) |
| Greenland Sea, Arctic | 17 | VRA | 7 | MC | (Boras et al., 2010b) |
| North | 9 | VRA | - | - | (Winter et al., 2005) |
| North Sea | 28 | VRA | - | - | (Parada et al., 2008) |
| North Atlantic | 26 | VRA | - | - | (Rowe et al., 2008) |
| North Atlantic | 18 | VRA | 10 | MC | (De Corte et al., 2012) |
| North Atlantic | 9 | VRA | 9 | MC | (Muck et al., 2014) |
| North Atlantic | 10 | VRA | - | - | (Winter et al., 2018) |
| Chesapeake Bay | 7 | VRA | - | - | (Winget and Wommack, 2009) |
| Tropical Atlantic | 9 | VRA | 8 | MC | (De Corte et al., 2010) |
| Baltic Sea | 6 | VRA | - | - | (Holmfeldt et al., 2010) |
| Baltic Sea | 9 | VRA | - | - | (Kostner et al., 2017) |
| Baltic Sea | 4 | VRA | - | - | (Šulčius et al., 2018) |
| Mediterranean | 24 | VRA | 11 | MC | (Boras et al., 2009) |
| Adriatic (Mediterranean) | 4 | VRA | - | - | (Bongiorni et al., 2005) |
| Mediterranean | 9 | VRA | - | - | (Fonda Umani et al., 2010) |

| | | | | | |
|---|---|---|---|---|---|
| Mediterranean | 1 | VRA | - | - | (Motegi et al., 2014) |
| Mediterranean | 21 | VRA | - | - | (Ordulj et al., 2017) |
| Cochin Estuary (Indian) | 35 | RIA | - | - | (Parvathi et al., 2013) |
| Cochin Estuary (Indian)) | 20 | VRA | - | - | (Jasna et al., 2017) |
| Indian Ocean | 10 | VRA | - | - | (Parvathi et al., 2018) |
| Bering Sea | 12 | FPB | - | - | (Steward et al., 1996) |
| Santa Monica Bay | 7 | FLVT | - | - | (Noble and Fuhrman, 2000) |
| Subtropics Atlantic, Pacific, Indian Ocean | 30 | VRA | 27 | MC | (Lara et al., 2017) |
| North Pacific (HOT) | 4 | VRA | - | - | (Brum, 2005) |
| North Pacific | 9 | VRA | - | - | (Hewson and Fuhrman, 2007) |
| North Pacific | 18 | VRA | - | - | (Gainer et al., 2017) |
| South Pacific | 16 | VRA | - | - | (Matteson et al., 2012) |
| Mariana Trench | 6 | VRA | - | - | (Li et al., 2018) |
| Western Pacific | 38 | VRA | - | - | (Li et al., 2014 |
| Japan Sea | 12 | FPB | - | - | (Hwang and Cho, 2002) |
| Masan Bay | 24 | FPB | - | - | (Choi et al., 2003) |
| Bach Dang Estuary (North Pacific) | 7 | VRA | - | - | (Bettarel et al., 2011a) |
| South China Sea | 13 | VRA | - | - | (Chen et al., 2011) |
| South China Sea | 5 | FLVT | - | - | (Nguyen-Kim et al., 2015) |
| Jiulong Estuary (North Pacific) | 27 | VRA | 27 | MC | (Chen et al., 2019) |

| | | | | | |
|---|---|---|---|---|---|
| Southwestern Pacific | 9 | VRA | - | - | (Rowe et al., 2012) |
| Southern Ocean (Indian sector) | 14 | VRA | - | - | (Evans et al., 2009) |
| Southern Ocean (Indian sector) | 10 | VRA | - | - | (Malits et al., 2014) |
| Southern Ocean (Pacific sector) | 16 | VRA | 8 | MC | (Vaque et al., 2017) |
| Southern Ocean (Atlantic sector) | 5 | VRA | - | - | (Weinbauer et al., 2009) |
| Southern Ocean (Atlantic sector) | 33 | VRA | 12 | MC | (Evans and Brussaard, 2012) |

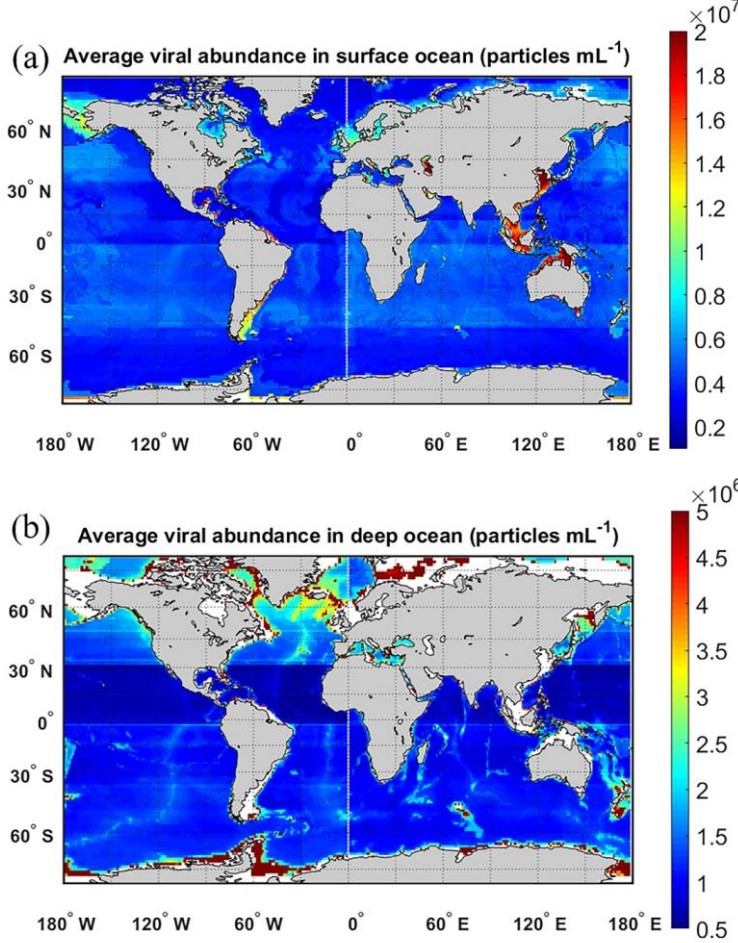

**Appendix Figure A1. Viral abundance projected using the random forest, showing average in (a) surface 200 m and (b) deep (>200 m) oceans.**

**Author contributions.**

RZ and YWL conceived and designed structure of database and mathematical analyses of the data. LX, WW, LC, XC and YH collected the data and described the metadata. LX, NJ, RZ and YWL conducted quality control and analyses of the data. LX, RZ and YWL led the writing of the paper, with contribution from all the co-authors.

**Competing interests.**

The authors declare that they have no conflict of interest.

**Acknowledgements.**

We would like to thank all the scientists and crews who collected the historical data. This work was funded by National Natural Science Foundation of China (91951209, 41890802, 42076153) and the MEL internal research fund (MELRI2003).

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

**Table 1. Number of accessory viral, host and oceanographic parameters associated with each of the core viral parameters (VA, lytic and lysogenic VP).**

| | VA ($n$=10,931) | Lytic VP ($n$=608) | Lysogenic VP ($n$=119) |
|---|---|---|---|
| *Accessory viral parameters* | | | |
| Frequency of lytic infection | 405 | 142 | 36 |
| Frequency of lysogenic infection | 227 | 96 | 36 |
| Viral decay rate | 83 | 65 | 27 |
| Burst size | 206 | 96 | - |
| Virus-mediated bacterial mortality | 46 | 46 | 19 |
| % cells lysed | 55 | 53 | 17 |
| *Accessory host parameters* | | | |
| Prokaryotic abundance | 10,846 | 598 | 119 |
| Prokaryotic productivity | 2,425 | 352 | 91 |
| Flagellate abundance | 411 | 44 | 7 |
| Picoeukaryotic abundance | 1,554 | 68 | 15 |
| *Synechococcus* abundance | 1,700 | 80 | 42 |
| *Prochlorococcus* abundance | 1,567 | 73 | 42 |
| *Accessory oceanographic parameters* | | | |
| Chlorophyll *a* | 3,949 | 244 | 71 |
| Temperature | 6,253 | 399 | 119 |
| Salinity | 6,360 | 370 | 85 |
| Oxygen | 4,930 | 82 | 46 |
| Nitrate | 2,707 | 91 | 46 |
| Phosphate | 3,153 | 144 | 51 |
| Silicate | 2,638 | 106 | 36 |
| pH | 96 | 47 | - |
| Light intensity | 35 | 35 | - |
| Dissolved organic carbon | 97 | 13 | - |

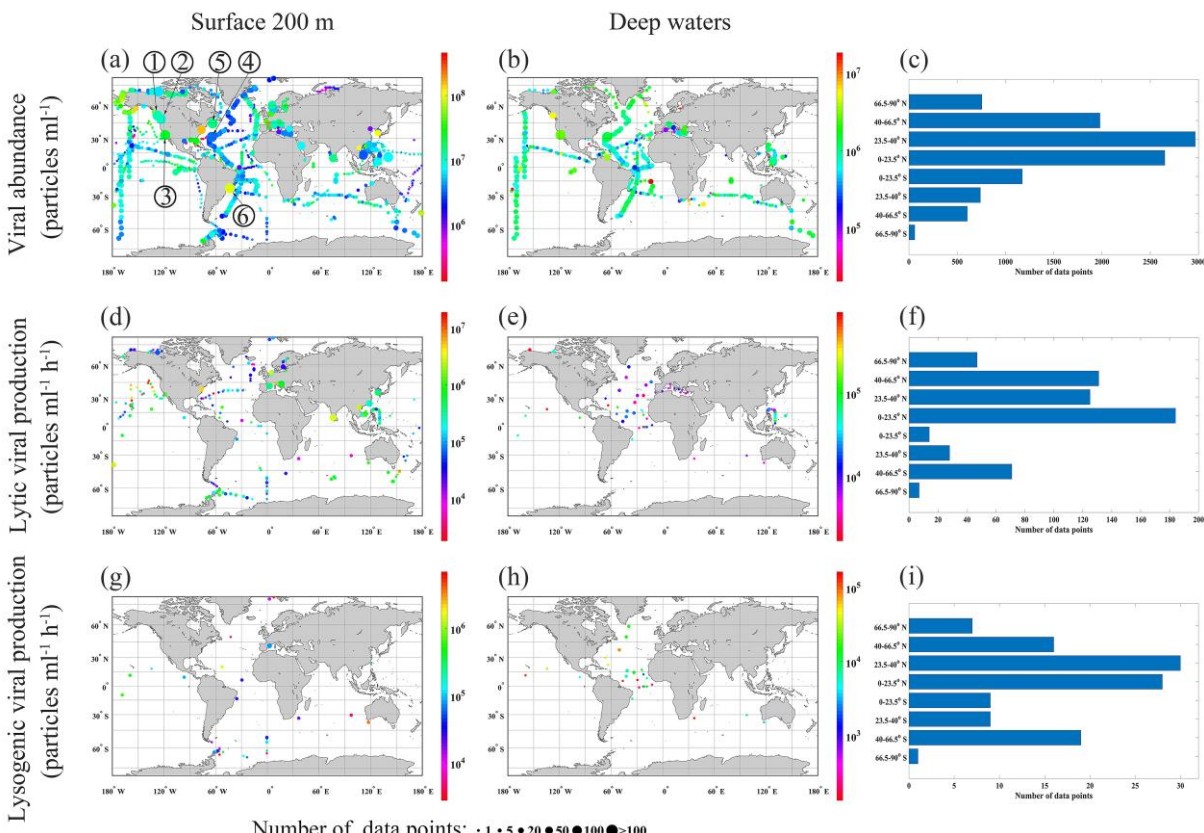

**Figure 1. Collected viral abundance (a, b), lytic viral production (d, f) and lysogenic viral production (g, h) in surface (≤200 m) (a, d and g) and in deep (>200 m) waters (b, e and h), binned on 1 °×1 °grids. Color of each grid codes the mean value of the parameters, and the size of the circles represents number of samples in each bin. Also shown the number of samples in latitudinal bands (c, f and i). Numbers in (a) represent long-term time-series observations of viral abundance: 1, Rivers Inlet; 2: Saanich Inlet; 3: San Pedro Ocean Time Series Microbial Observatory; 4: Bedford Basin Monitor; 5: Bermuda Atlantic Time-series Study and 6: Guanabara Bay.**

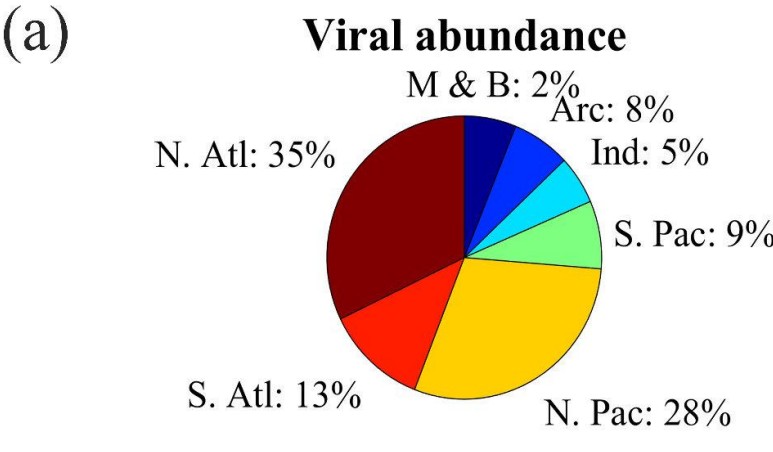

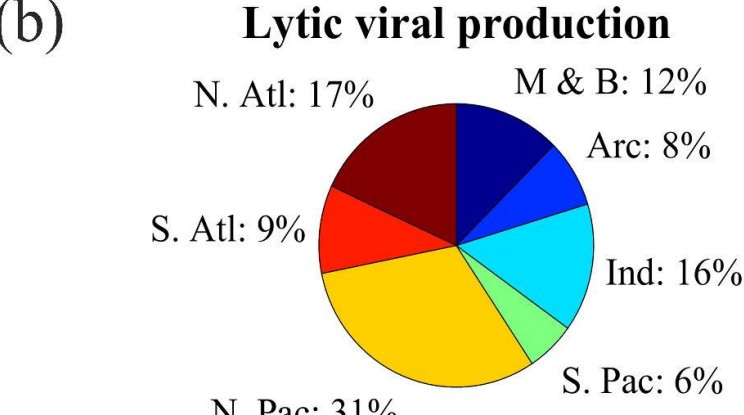

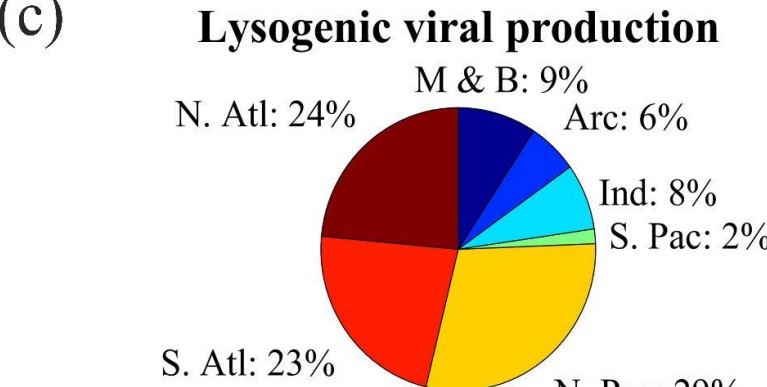

**Figure 2. The fraction of (a) viral abundance, (b) lytic viral production and (c) lysogenic viral production data points in different oceans (N. Atl: North Atlantic, S. Atl: South Atlantic, N. Pac: North Pacific, S. Pac: South Pacific, Ind: Indian ocean, Arc: Arctic ocean, M & B: Mediterranean Sea and Baltic Sea).**


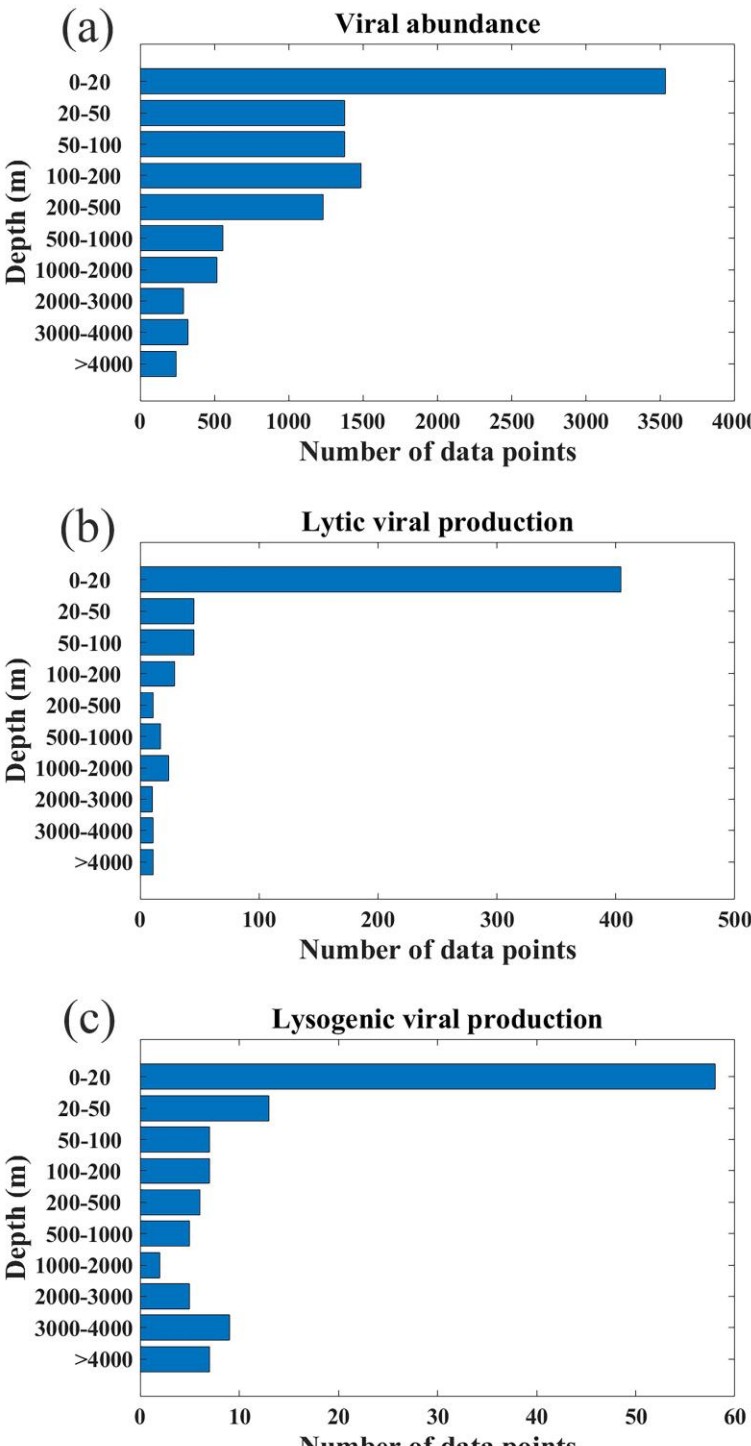


**Figure 3. Depth distribution of the data for (a) viral abundance, (b) lytic viral production, and (c) lysogenic viral production.**

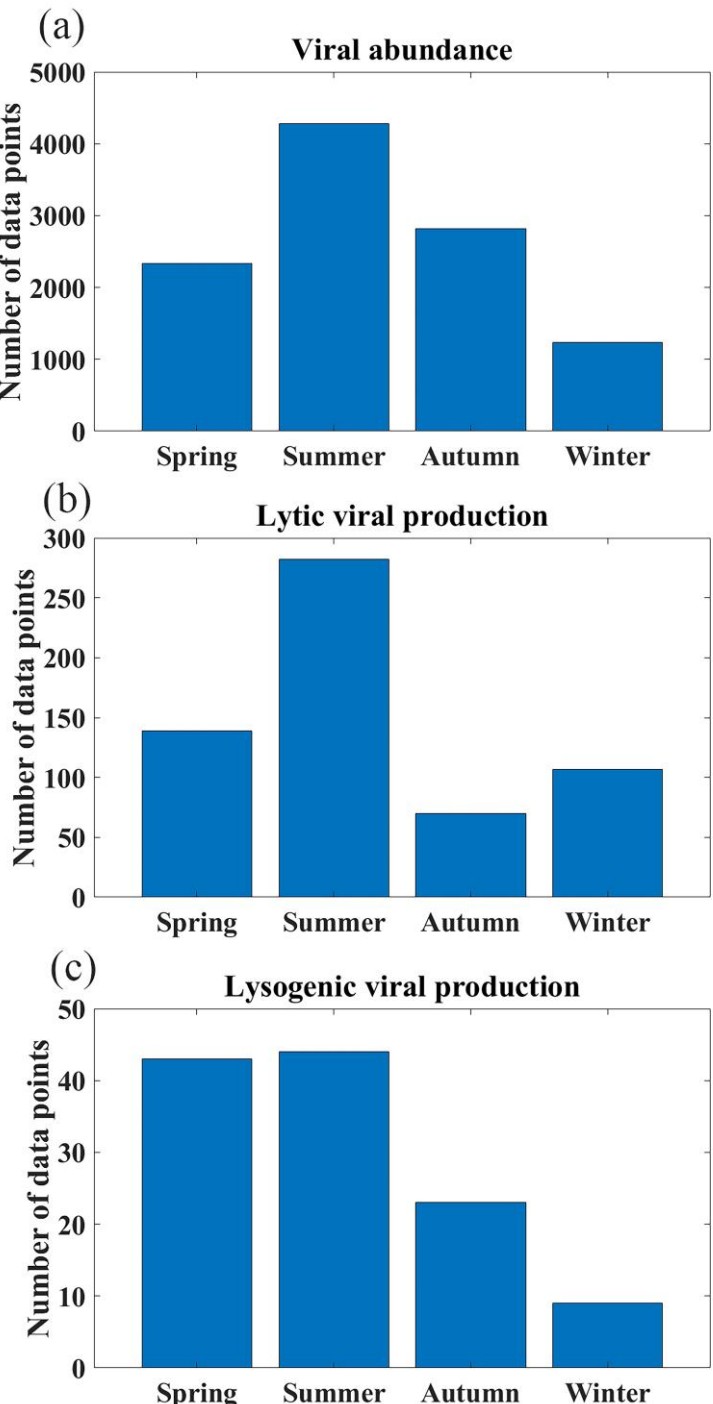

**Figure 4. Seasonal distributions of number of samples for (a) viral abundance, (b) lytic viral production and (c) lysogenic viral production.**

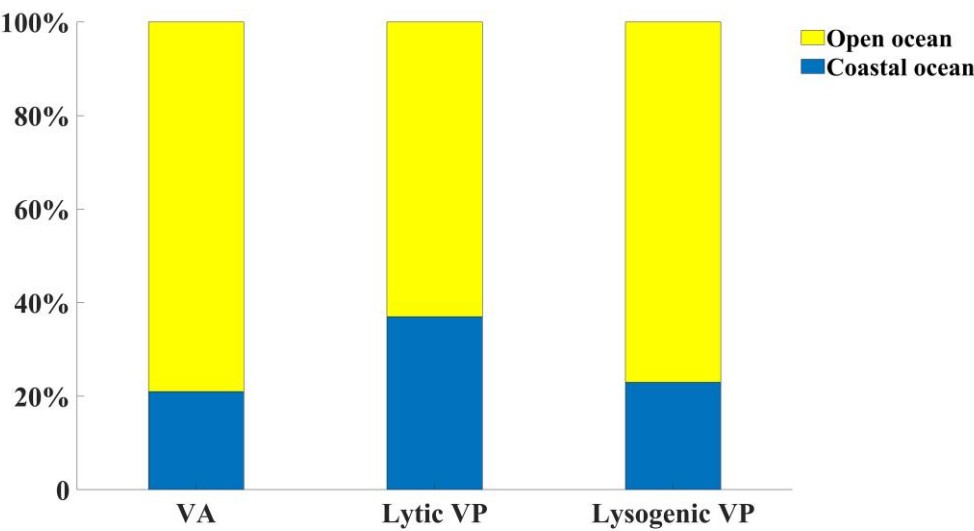

**Figure 5. The fraction of viral abundance (VA), lytic viral production (VP) and lysogenic VP data points in coastal versus in open oceans.**


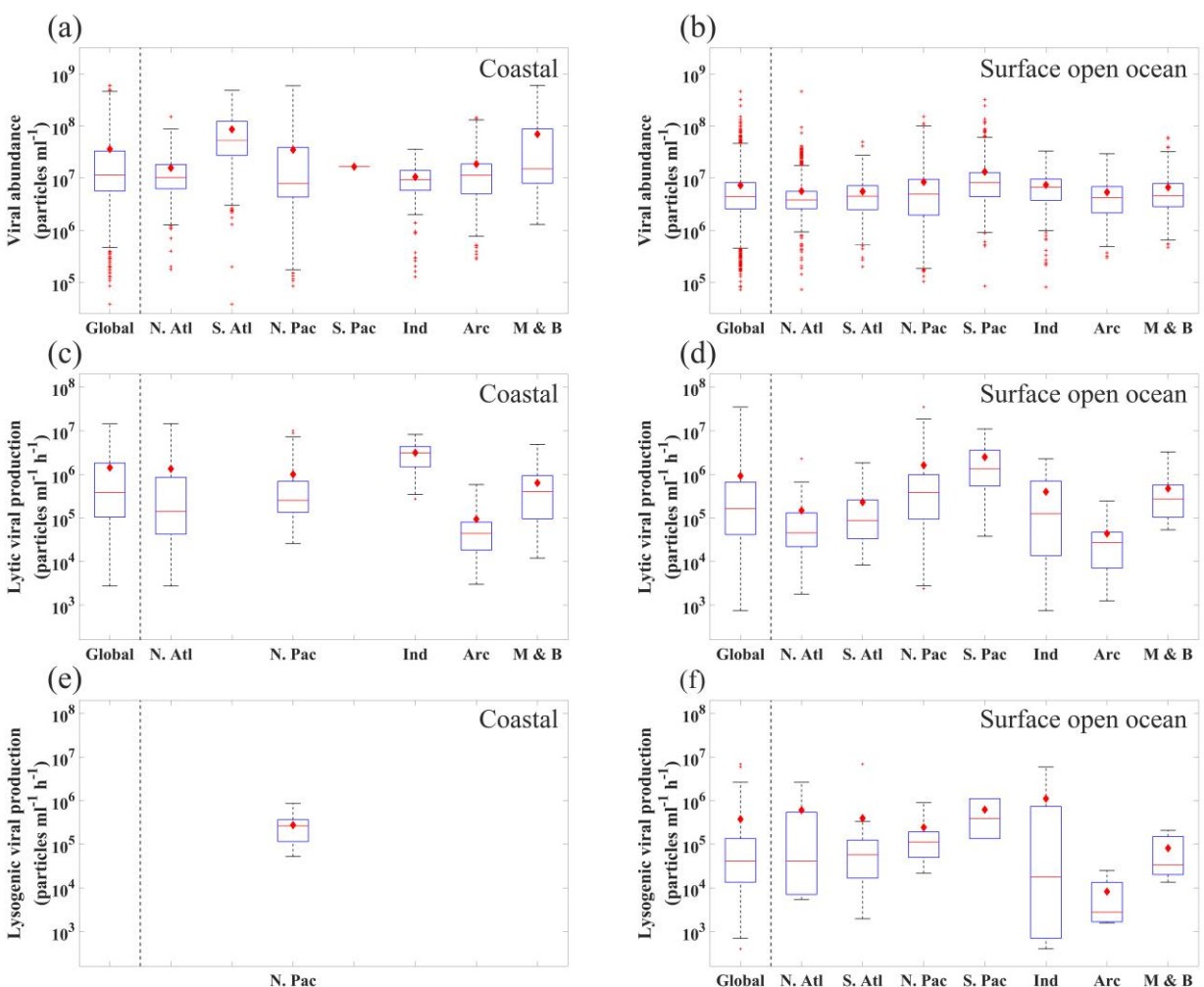

**Figure 6. The range of (a, d) viral abundance, (b, e) lytic viral production, and (c, f) lysogenic viral production in the different oceans (N. Atl: North Atlantic, S. Atl: South Atlantic, N. Pac: North Pacific, S. Pac: South Pacific, Ind: Indian ocean, Arc: Arctic ocean, M & B: Mediterranean Sea and Baltic Sea), grouping in coastal ocean (a–c) and open ocean samples in surface 200 m (d–f). All data are shown in logarithmic scales. The red diamonds mark the mean value. The central red lines indicate the median, and the bottom and top edges of the box indicate the 25th and 75th percentiles of the data. Error bars extend to the 5th and 95th percentiles and the remaining outliers are marked with red plus signs.**

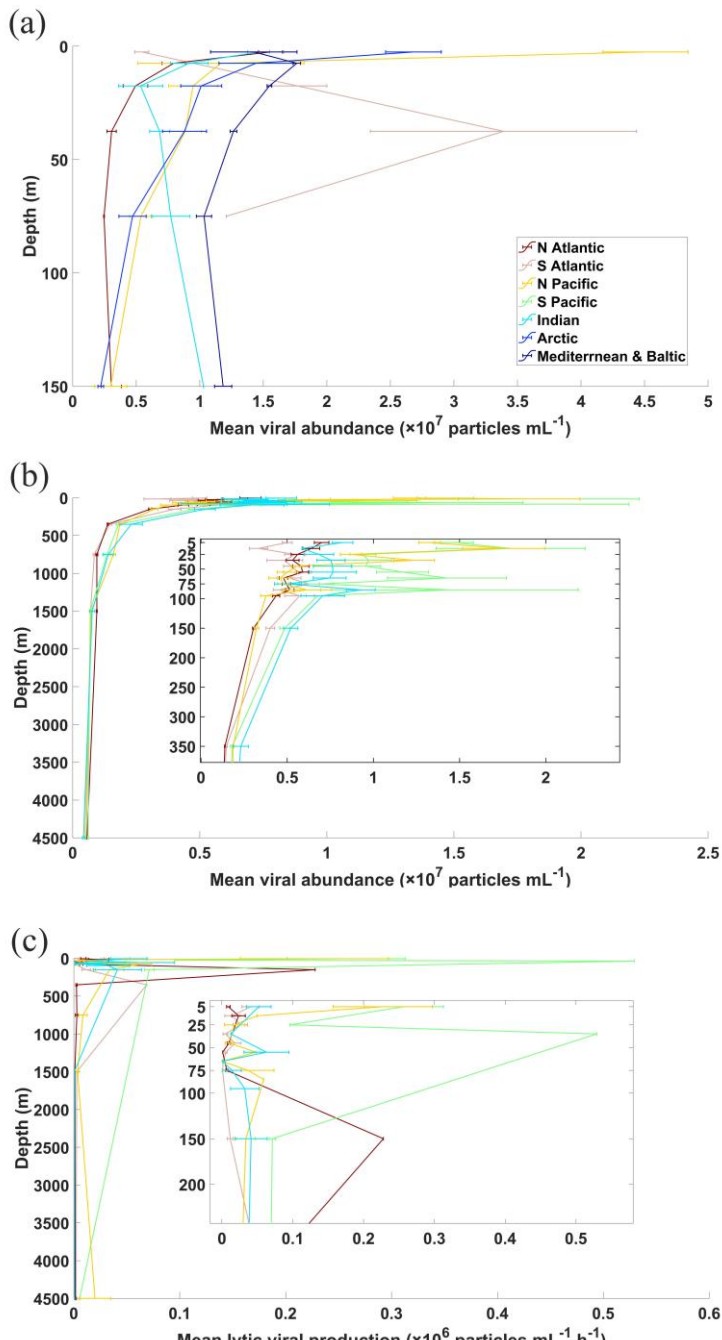

**Figure 7. Vertical profiles of average viral abundance of each ocean basin in (a) coastal and (b) open ocean waters. Also shown the vertical profiles for open water lytic viral production (c), while those for coastal samples were not constructed because of limited data points. Error bars represent one standard error of the mean.**

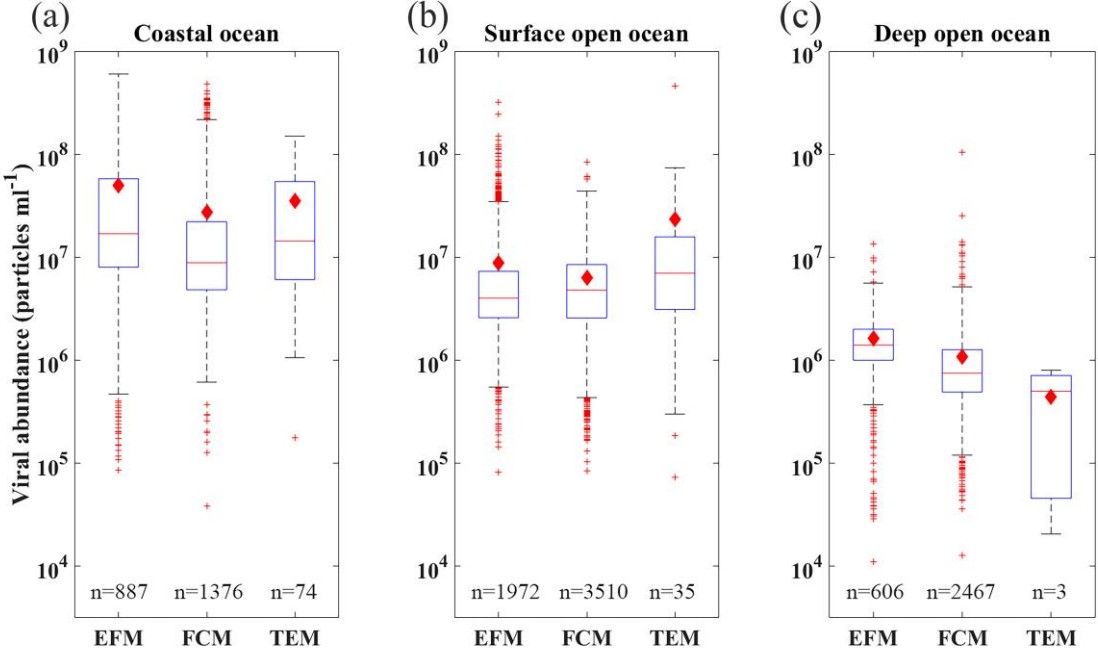

**Figure 8. Box plots of distribution of viral abundance using different measurement methods (EFM, FCM and TEM, see text for**
**more details) in coastal (a), surface open ocean (b) and deep open ocean (c) waters. See caption of Figure 6 for details of lines and**
**symbols of the box plots.**


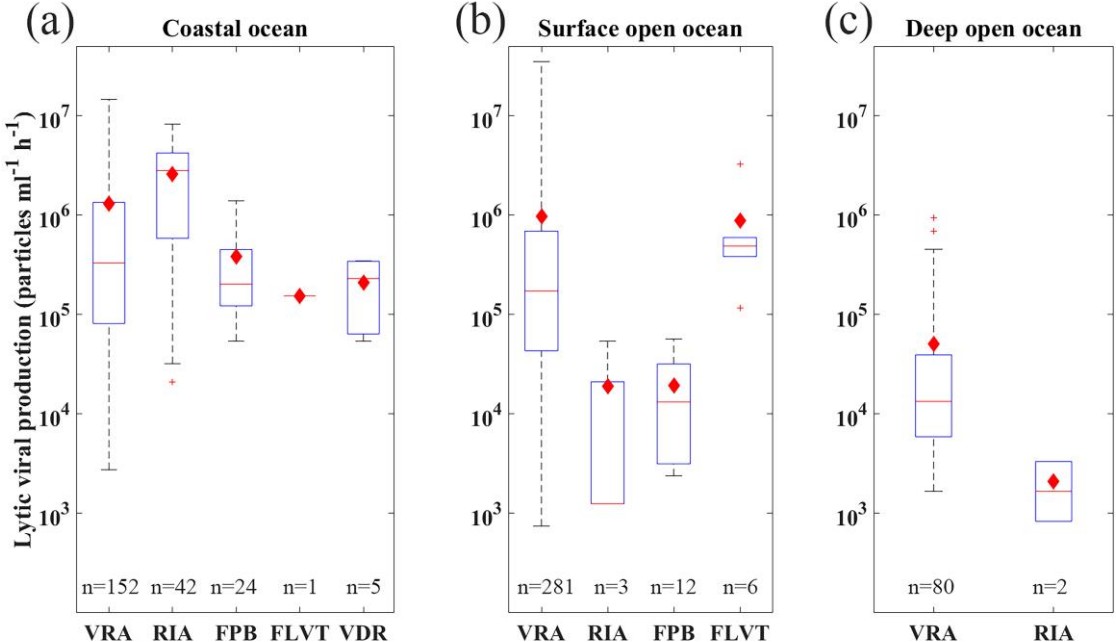

**Figure 9. Comparison of lytic viral production using different measurement methods including VRA, RIA, FPB, FLVT and VDR**

**(FPB: calculated by multiplying fraction of viral infected cells, prokaryotic production and burst size; RIA: radioactive incorporation approach; FLVT: fluorescently labelled viral tracers method; VRA: virus reduction approach; VDR: estimated by viral decay rates. See text for details) in coastal (a), surface open ocean (b) and deep open ocean (c) waters. See caption of Figure 6 for details of lines and symbols of the box plots.**