# Peer review of "A global viral oceanography database (gVOD)"

_Earth System Science Data, 2020_

## Referee Comment (RC1) · Anonymous Referee #1 · 28 Oct 2020

The authors of the manuscript construct the first global viral oceanography database containing viral abundance (VA) and viral production (VP) data with host and environmental parameters which are collected from published papers. Based on the database, the authors estimate the total viral number and biomass in the global ocean, and compare the technological bias of different methods of data generation. This database can be valuable for field and modeling studies in marine ecology, biogeochemistry and other areas of oceanography. This manuscript needs to be improved before acceptance for publication.

Specific Comments: 1) Line 175-180: "VA counted by the three methods distribute in similar ranges and do not show systematic difference". In the database, each data point is only used one of three methods to obtain the viral population data. So, how do the authors compare the difference of three methods? Are there some samples which use two of the three methods? 2) Line 196-206: It is same as above. Five methods

are used to generate the VP data. But no sample is used more than two of the five methods. The authors should describe how they compare the differences of these methods. 3) In Figure 2, it may be better that the figure of each fraction is shown.

---

## Referee Comment (RC2) · Anonymous Referee #2 · 30 Dec 2020

Review for "A global viral oceanography database (gVOD)" Xie et al., Earth System Science Data, Dec 2020

Summary

The authors have gathered an extensive dataset describing global oceanographic virus abundance and productivity, along with other oceanographic and environmental data (salinity, temperature, etc.). Virus abundance data using three different methods (transmission electron microscopy, flow cytometry, and epifluorescence microscopy) are included in the dataset. Furthermore, data describing lytic virus production using five complementary methods, and lysogenic production using mitomycin C treatments, are included. The authors provide a summary of the geographic patterns evident in their data and use two complementary statistical models to infer global virus abundance and biomass, the latter utilizing a model relating virus carbon content to capsid size. Their

global estimates are consistent with, and complementary to, prior estimates of global ocean virus abundance. The authors also leverage their new dataset to assess whether complementary methods to infer abundance produce consistent results in similar environments. They find, reassuringly, that different methods to measure virus abundance produce similar results.

Main comments

This manuscript has been extremely carefully put-together. I really appreciated the concise explanation of different techniques used, and the clarity with which the results are reported. Clearly, a lot of effort has gone into this work. I enjoyed seeing the methodology used to infer global virus abundance (I always see the 10ˆ30 number banded around, without knowing how those estimates are reached). I also appreciate that effort has been made to quantify uncertainty in all of their estimates, and to evaluate the consistency of different techniques. Overall, this manuscript is a timely and necessary contribution.

Specific comments

Overall, the language is very clear, there are only very minor English language issues. I pointed out a couple, but it could be useful to have one final sweep through to look for language edits.

Line 8: "Virioplankton are a key component of the marine biosphere" (suggested change)

Line 9: "They also contribute greatly to nutrient cycles/cycling" (suggested edit)

Line 21: The link to the database is no longer valid, I understand it was temporary and wouldn't have been an issue if I had done my review earlier. Noting it will need to be updated nevertheless.

Line 88: "For notational simplicity, " (suggested edit). Also, maybe clarify what FPB stands for? Is it Fraction-Prokaryote-Burst?

Line 122: consider providing your code used for the modeling, either in a public repository, or as a supplement to the article

Line 210: It looks like the range for lysogenic production goes negative? Is this an artefact or something real? These negative ranges sometimes happen when you take a standard deviation of data that are heavily positive-skew (often the case with biological data). Log-transforming can help (although I see you have done this elsewhere). May be worth commenting on this point to clarify if it is an artefact or something real.

Line 214: update link

---

## Author Comment (AC1) · 25 Jan 2021

We thank both reviewers for their very constructive comments and suggestions that greatly helped us improve our manuscript. We have responded (in blue fonts) to the comments point by point and revised the manuscript accordingly.

**To Reviewer 1:**

The authors of the manuscript construct the first global viral oceanography database containing viral abundance (VA) and viral production (VP) data with host and environmental parameters which are collected from published papers. Based on the database, the authors estimate the total viral number and biomass in the global ocean, and compare the technological bias of different methods of data generation. This database can be valuable for field and modeling studies in marine ecology, biogeochemistry and other areas of oceanography. This manuscript needs to be improved before accep-

tance for publication.

**Response:** We would thank the reviewer for her/his careful reading and the positive comments on our work. We have improved the quality of the manuscript according to your valuable suggestions. Please see the detailed response below.

Specific comments:
1) Line 175-180: "VA counted by the three methods distribute in similar ranges and do not show systematic difference". In the database, each data point is only used one of three methods to obtain the viral population data. So, how do the authors compare the difference of three methods? Are there some samples which use two of the three methods?

**Response:** Sorry for the confusion. Yes, the reviewer was correct that in our database, the viral abundance of each sample was determined by only one of the three methods. Thus, we were not trying to directly compare three methods which required parallel experiments for same samples. Please note that there were some classic and excellent papers directly compared these methods such as Brussaard et al., 2010; Marie et al., 1999; Payet and Suttle, 2008 but their data can not be included in our database due to the lack of crucial information (e.g., sampling position, exact number, etc.). Instead, our analysis here supported previous technical comparison studies that viral counts with different methods were consistent in similar environments (although for different samples). Nevertheless, our database provides references for methodological comparison in the future.

We then added the discussion in the revised manuscript (**line 175-181**): "Previous studies have showed that the VA counted using FCM, which became more popular in studies after 2014 (Table S1), had a strong correlation with those using EFM (Brussaard et al., 2010; Marie et al., 1999; Payet and Suttle, 2008). Our data demonstrated that the VA obtained by FCM and EFM methods has consistent results in similar environments. For deep open ocean samples, VA using TEM are substantially lower

than those using the other two methods (Fig. 8). But considering much fewer VA data points using TEM than others (Fig. 8 Table S1), we cannot conclude TEM substantially underestimated VA in the deep water samples. Nevertheless, our database provides references for methodological comparison in the future."

References:

Brussaard, C. P. D., Payet, J. P., Winter, C., and Weinbauer, M. G.: Quantification of aquatic viruses by flow cytometry. In: Manual of aquatic viral ecology, https://doi.org/10.4319/mave.2010.978-0-9845591-0-7.102, 2010.
Marie, D., Brussaard, C. P. D., Thyrhaug, R., Bratbak, G., and Vaulot, D.: Enumeration of marine viruses in culture and natural samples by flow cytometry, Appl Environ Microbiol, 65, 45-52, 10.1128/AEM.65.1.45-52.1999, 1999.
Payet, J. P. and Suttle, C. A.: Physical and biological correlates of virus dynamics in the southern Beaufort Sea and Amundsen Gulf, J Marine Syst, 74, 933−945, https://doi.org/10.1016/j.jmarsys.2007.11.002, 2008.

2) Line 196-206: It is same as above. Five methods are used to generate the VP data. But no sample is used more than two of the five methods. The authors should describe how they compare the differences of these methods.

**Response:** The reviewer was correct that only one or two methods were used to determine the lytic VP of each sample in our database. As that for VA, we had not directly compared these five methods. However, in similar environments, our statistics showed that the lytic VP rates determined by FLVT and VRA were higher than those measured by RIA. This also provided certain support for previous methods comparison studies (Helton et al., 2005; Karuza et al., 2010; Rastelli et al., 2016).

we added the discussion in the revised manuscript to avoid confusion (**Line 198-209**): "Several studies have tried to compare different approaches estimating the lytic VP, revealing that the VRA method was more reliable and less laborious, compared to

the probable overestimation by FLVT approach and the potential underestimation by RIA method, though such comparisons were mainly constrained to the coastal ocean (Helton et al., 2005; Karuza et al., 2010; Rastelli et al., 2016; Winget et al., 2005). Additionally, although a meaningful comparison of reported lytic VP values between disparate marine ecosystems is complicated by the inherent variability among approaches, the lytic VP rates in this database might provide a tentative global-scale insight into methodological comparison. Our statistics showed that, in similar environments, the lytic VP rates determined by FLVT and VRA were higher than those measured by RIA. For coastal samples, such difference among methods was not obvious (Fig. 9). However, due to the limited number of samples using the methods other than VRA (Fig. 9 and Table S2), we did not have adequate data to tell if the difference in VP was caused by the measurement methods, or the randomness of the samples. Hence, more measurements of lytic VP using multiple approaches simultaneously will be certainly needed to better evaluate the differences among them."

References:
Helton, R. R., Cottrell, M. T., Kirchman, D. L., and Wommack, K. E.: Evaluation of incubation-based methods for estimating virioplankton production in estuaries, Aquatic Microbial Ecology, 41, 209-219, DOI 10.3354/ame041209, 2005.
Karuza, A., Del Negro, P., Crevatin, E., and Fonda Umani, S.: Viral production in the Gulf of Trieste (Northern Adriatic Sea): Preliminary results using different methodological approaches, J Exp Mar Biol Ecol, 383, 96−104, https://doi.org/10.1016/j.jembe.2009.12.003, 2010.
Rastelli, E., Dell'Anno, A., Corinaldesi, C., Middelboe, M., Noble, R. T., and Danovaro, R.: Quantification of viral and prokaryotic production rates in benthic ecosystems: A methods comparison, Front Microbiol, 7, 1501, https://doi.org/10.3389/fmicb.2016.01501, 2016.

3) In Figure 2, it may be better that the figure of each fraction is shown.

**Response:** Thank you for the specific suggestion. We added the percentage of each fraction in the figure (see figure 2 in the revised manuscript)

---

## Author Comment (AC2) · 25 Jan 2021

We thank both reviewers for their very constructive comments and suggestions that greatly helped us improve our manuscript. We have responded (in blue fonts) to the comments point by point and revised the manuscript accordingly.

**To Reviewer 2:**

Summary:
The authors have gathered an extensive dataset describing global oceanographic virus abundance and productivity, along with other oceanographic and environmental data (salinity, temperature, etc.). Virus abundance data using three different methods (transmission electron microscopy, flow cytometry, and epifluorescence microscopy) are included in the dataset. Furthermore, data describing lytic virus production using five complementary methods, and lysogenic production using Mitomycin C treatments, are

included. The authors provide a summary of the geographic patterns evident in their data and use two complementary statistical models to infer global virus abundance and biomass, the latter utilizing a model relating virus carbon content to capsid size. Their global estimates are consistent with, and complementary to, prior estimates of global ocean virus abundance. The authors also leverage their new dataset to assess whether complementary methods to infer abundance produce consistent results in similar environments. They find, reassuringly, that different methods to measure virus abundance produce similar results.

Main comments

This manuscript has been extremely carefully put-together. I really appreciated the concise explanation of different techniques used, and the clarity with which the results are reported. Clearly, a lot of effort has gone into this work. I enjoyed seeing the methodology used to infer global virus abundance (I always see the 10Ȩ̈30 number banded around, without knowing how those estimates are reached). I also appreciate that effort has been made to quantify uncertainty in all of their estimates, and to evaluate the consistency of different techniques. Overall, this manuscript is a timely and necessary contribution.

**Response:** We thank the reviewer for her/his support and very useful comments to our manuscript. We have revised the manuscript accordingly. The detailed responses are listed below.

Specific comments:

Overall, the language is very clear, there are only very minor English language issues. I pointed out a couple, but it could be useful to have one final sweep through to look for language edits.

Line 8: "Virioplankton are a key component of the marine biosphere" (suggested change)

Line 9: "They also contribute greatly to nutrient cycles/cycling" (suggested edit)

**Response:** Thanks for your reminding. We checked the language thoroughly and numerous edits, including the two pointed out by the reviewer, were made in the revised manuscript.

Line 21: The link to the database is no longer valid, I understand it was temporary and wouldn't have been an issue if I had done my review earlier. Noting it will need to be updated nevertheless.

**Response:** Thanks for your reminding. The data repository, PANGAEA, has granted a persistent DOI link to the database https://doi.org/10.1594/PANGAEA.915758, although it takes up to 30 days to become valid once the DOI registration process is completed. The link has been updated (**Line 21**).

Line 88: "For notational simplicity," (suggested edit). Also, maybe clarify what FPB stands for? Is it Fraction-Prokaryote-Burst?

**Response:** Yes, the FPB stands for Fraction-Prokaryote-Burst. We changed "For simplifying reason" to "For notational simplicity" and added the explanation of FPB in the revised manuscript (**Line 89 and 90**).

Line 122: consider providing your code used for the modeling, either in a public repository, or as a supplement to the article.

**Response:** Thanks for your suggestion. The code (a MATLAB script) will be attached as a supplement material to the article.

Line 210: It looks like the range for lysogenic production goes negative? Is this an artefact or something real? These negative ranges sometimes happen when you take a standard deviation of data that are heavily positive-skew (often the case with biological data). Log-transforming can help (although I see you have done this elsewhere). May be worth commenting on this point to clarify if it is an artefact or something real.

**Response:** The reviewer was correct that the standard deviation higher than the mean is an artefact because of some large positive data. We decide to use mean of the original data (i.e. mathematic mean), instead of that of log-transformed data (i.e. the geometric mean), and the associated standard deviations, to facilitate direct comparing with numbers reported by other studies.

We added the range of the data for clarifying: "The overall mean and standard deviation of lytic VP in the global ocean were $9.87(\pm24.16) \times 105$ particles ml-1 h-1 (ranging in $0.00746 \times 105 - 350 \times 105$)." (**Line 190**)
"The overall lysogenic VP in the global ocean is $2.53(\pm8.64) \times 105$ particles ml-1 h-1 (ranging in $0.00132 \times 105 - 68.8 \times 105$)." (**Line 212**)

Line 214: update link

**Response:** The link has been updated (**Line 216**).

---

## Author Response (AR2)

**Response to Editor's** comments to the Author:

Thank for using ESSD, for many positive and helpful responses to reviewer comment, and for extreme patience.

Because ESSD publishes descriptions of many open active databases (e.g. soliciting new contributions from the community at any time), please clarify that gVOD represents a static one-time (extensive!) compilation. Impressive but not open nor actively seeking new contributions. Perhaps mention some future update?

**Response:** Thanks for your suggestion. We clarified in the revised manuscript "The gVOD is a compilation of all the available data, to our best knowledge, by 2019. We plan to update the database every 5 years." (Line **74**)

Because Copernicus (publisher) does not archive supplements, some of this supplementary material might in the future become separated from the VOD description? Consider options of what to include with manuscript as appendices (e.g. which will get archived), what to include with the gVOD product at PANGAEA, etc. Supplement okay so long as you accept what you might lose in the future?

**Response:** Thanks for reminding. All the supplementary tables and figures in previous supplementary have been integrated as appendices of the revised manuscript. We also decided to keep the MATLAB code package for calculating the total number of viruses in supplementary materials, as the package includes associated MATLAB datasets that cannot be appended in the manuscript, and meanwhile remind the readers they can also request the codes from us: "The MATLAB codes for calculating the total number of viruses can be found in the supplementary materials or be obtained by requesting the corresponding authors." (Line **220**)